# The Perception of Rural Medical Students Regarding the Future of General Medicine: A Thematic Analysis

**DOI:** 10.3390/healthcare9101256

**Published:** 2021-09-24

**Authors:** Kasumi Nishikawa, Ryuichi Ohta, Chiaki Sano

**Affiliations:** 1Faculty of Medicine, Shimane University, 89-1 Enya cho, Izumo 693-8501, Japan; k.nishikawa0324@gmail.com; 2Community Care, Unnan City Hospital, 699-1221 96-1 Iida, Daito-cho, Unnan 699-1221, Japan; 3Department of Community Medicine Management, Faculty of Medicine, Shimane University, 89-1 Enya cho, Izumo 693-8501, Japan; sanochi@med.shimane-u.ac.jp

**Keywords:** medical student, perception, medical education, general medicine, family medicine, rural, Japan

## Abstract

Although the demand for general physicians has increased in Japan because of its aging population, medical universities primarily provide organ-based education; thus, medical students do not receive sufficient general medical education. The number of residents focusing on general medicine remains low; therefore, to understand the present situation regarding general medicine education, we attempted to clarify the views of medical students and the factors influencing them. In this qualitative study, semi-structured interviews were conducted in 12 medical students at Shimane University, and the results were analyzed through thematic analysis. The results indicated the emergence of three themes and 14 concepts. The three overarching themes were as follows: hopes for the field of general medicine, gaps between ideal and reality of general medicine, and factors affecting students’ motivation for specialization in general medicine. Medical students had a positive impression of general medicine and believed that it has potential for further development; however, they felt a gap between their ideals and reality (i.e., unclear expertise). Factors creating this gap included poorly developed education and medical policies. We need to restructure general medicine education based on the participants’ perceptions by establishing collaborative curricula between universities and community hospitals and by increasing students’ exposure to general medicine.

## 1. Introduction

As the aging population rapidly increases worldwide, the demand for primary care and family medicine is expected to increase [1]. Different countries have different names for physicians involved in primary care, such as “family physician”, “primary care physician”, and “general practitioner”; in Japan, they are referred to as “general physicians”. General physicians possess a wide variety of medical knowledge, can collaborate with various healthcare professionals, and can navigate complex medical problems [2,3]. Additionally, they can also help alleviate the shortage of physicians in rural areas and reduce regional disparities [2,4]. Consequently, the Japanese Medical Specialty Board officially added “general medicine” as a 19th specialty, thereby establishing its own training program [5]; however, only 2.4% (222/9082) of physicians registered in the program in 2020 [5].

The curriculum in Japanese medical schools—established by the Japanese Ministry of Health, Labor, and Welfare—requires that medical students be taught general medicine, comprehensive care, and primary care [6]. However, a previous study showed that medical students primarily studied organ-based medicine and did not have enough opportunities to learn concerning general medicine [7], thereby diminishing their interest in the field. The educational system innovations, such as increasing medical students’ exposure to general medicine, may increase the number of students interested in it.

In the United States (US), Canada, Australia, and even the United Kingdom (UK), there is a shortage of primary care physicians, and the popularity of primary care among medical students has declined [8]. A previous study in Japan revealed that medical students tended to aim for internal medicine (65.9%), followed by general practice (32.3%), pediatrics (29.4%), surgery (27.2%), and emergency medicine (19.3%) [9]. Although the percentage of students who are interested in general medicine is not so small [9], the number of physicians with expertise in this field is limited [5]. Another study revealed that medical students who chose general practice preferred frequent patient communication, engaging in clinical diagnostic reasoning, and community-oriented practice [9]. Additionally, other studies have revealed that relevant experiences in clinical settings along with having general physician role models were important for increasing students’ interest in general medicine [10,11]. Based on the above, the research question of our study was: what prevents medical students from becoming general physicians?

It is important for the future of general medicine to clarify medical students’ current perceptions regarding general medicine. These were not fully investigated after the training program was established in Japan. There may be different perceptions of general medicine because of the differences in educational background. Previous studies in the UK and US have revealed that the main factors limiting primary care physician development were the reputation of primary care and the lack of primary care experiences in clinical practice [8,12]. The aims of this study were to investigate perceptions of general medicine, determine the factors affecting those perceptions, and propose solutions for promoting general medicine education. We accomplished these objectives by interviewing medical students at Shimane University in Japan. The results of our investigation may help improve and develop general medicine education in Japan and other countries.

## 2. Materials and Methods

### 2.1. Study Design

In this qualitative study, we conducted semi-structured interviews of medical students and analyzed the interviews using a thematic analysis. This allowed us to substantially investigate and identify the medical students’ views on general medicine.

### 2.2. Setting

This study was conducted in Shimane, which is a local prefecture in Japan with eight cities, five counties, 10 towns, and one village. The population of the prefecture is 665,702 (as of 1 January 2021), and the proportion of the population aged ≥65 years is 34.7% (as of 1 October 2020) [13]. Shimane University is the only medical school in the area, and similar to other universities, it has a selective admissions system, referred to as “Chiiki-Waku”. In the Chiiki-Waku system, students are obliged to work in the prefecture for a certain duration after graduation instead of paying back their student loans. Ten hospitals in Shimane, including university and community hospitals, have established training programs for general medicine [14]. There are 301.5 physicians per 100,000 people in Shimane (national average: 258.8 per 100,000 people as of 31 December 2018) [15], with 10 family medicine specialists, 27 certified physicians, and 26 instructors [16]. There are eight general physicians in training in Shimane (zero, three, and eight in 2018, 2019, and 2020, respectively) [17].

### 2.3. Participants

There were 696 medical students at Shimane University at the time of study [18]. We posted the information on social networking service which almost all of the students used and recruited those who were interested in general medicine. As of 2019 (before the COVID-19 pandemic), the participants had learned concerning general medicine based on the Shimane University Faculty of Medicine curriculum (Table 1) [19].

### 2.4. Data Collection

The first author of this study selected the participants who were expected to present their own opinions concerning general medicine in response to the research questions, and conducted the semi-structured interviews for this qualitative study. The author is a 5th-year student at Shimane University Faculty of Medicine who aims to be a general physician and who was educated by a general physician at Unnan City Hospital. One-on-one interviews lasting approximately 30 min were conducted with participants using a video conferencing tool. All the interviews were recorded, and the audio was transcribed verbatim. As a guided interview for the semi-structured interviews, we first asked all participants three questions: “what do you think of general medicine?”, “what prevents you from specializing in general medicine?”, and “what should be improved regarding general medicine?” Following these questions, the interviewer added questions and expanded the topics according to the answers to understand the participants’ real perceptions. The interviewer and the coauthor who is a general physician analyzed and discussed the data after each interview. Then, we added any unclear and newly emerging questions in addition to the three questions. We continued to collect and analyze the data until the data were saturated. We performed 12 interviews in total.

### 2.5. Analysis

We performed thematic analyses using the six steps proposed by Braun and Clarke [20]. In the first step, the first author and coauthor independently read the transcript to understand the content; in the second step, the transcript was coded. For open coding, we divided the transcripts into units and defined and labeled them. After the first author coded each transcript, we discussed and refined the coding for credibility. For axial coding, we found the relationships between the codes, organized and grouped them, and created a concept. In the third step, we identified the relationships and patterns between concepts and combined the codes into a theme. In the fourth step, the themes were reviewed and improved by the researchers. In the fifth step, each theme was named and defined. In the final step, we identified relationships between the themes, and wrote the results of our analysis. We alternated between data collection and analysis, confirming that no new codes had emerged, and finally created concepts and themes. After the 12th interview, theoretical saturation was confirmed.

### 2.6. Ethical Considerations

The participants were informed, in advance, that the data collected in this study would be used only for this study and not for any other purpose. Informed consent was obtained from all participants involved in the study. The Unnan City Hospital Clinical Ethics Committee approved this study (Approval Code: 0200027).

## 3. Results

Twelve medical students were interviewed, and the male:female ratio was 2:1 (i.e., eight male and four female individuals), representing the approximate male:female ratio of the Japanese medical students [21]. Of the 12 participants, there were three, five, two, and two 6th-year, 5th-year, 4th-year, and 3rd-year students, respectively.

Upon analysis, three themes (hopes for the field of general medicine, gaps between the ideal and reality of general medicine, and factors affecting students’ motivation for specialization in general medicine) and 14 concepts were emerged (Table 2). The participants had interest in general medicine and its future owing to the general medicine education they had acquired. However, they felt that there was a gap between the reality and ideals of general medicine concerning education and exposure to clinical situations; reality was considered the lack of effective education and exposure to clinical situations typical of general medicine. Furthermore, the ever-changing environments of health care and general medicine as well as the duty to work in rural contexts affected their motivation for general medicine specialization.

### 3.1. Hopes for the Field of General Medicine

#### 3.1.1. Community-Based Medicine

One advantage of general medicine that makes it a preferable option is its role in community-based family medicine. The participants described a desire to provide continuous health care to the local community, provide home healthcare, and collaborate with multiple healthcare professionals. The participants recognized that general physicians manage disease prevention and public health within the local community and that they provide patients with holistic care, based on the biopsychosocial model.

One of the participants stated: 


*“I have the impression that (general physicians) are very closely related to the community. I think that managing the entire community is very broad. General physicians cover a wide range [of services], from public health to management of various diseases”*
(Student 2).

The participants typically had a positive impression of general medicine that it is community-based and allows general physicians to provide continuous support and treatment for patients’ lives and illnesses. Additionally, the medical students recognized that general physicians provide disease prevention, public health guidance, and home healthcare, and manage community health according to local needs in collaboration with multiple healthcare professionals.

#### 3.1.2. Broad Scope of Practice

The participants felt that general physicians provide medical care for a wide range of symptoms and diseases (regardless of the field or patient age) and manage patients whose healthcare needs cannot be classified into organ-specific departments. Therefore, the medical students believed that general physicians could manage most of the local community’s medical problems, excluding highly specialized medical needs.

One of the participants stated: 


*“I think that general physicians can practice a rather wide range of medical specialties. However, they may not be able to manage particularly specialized medical problems. I feel that they have a very broad perspective”*
(Student 9).

The participants felt that general physicians had the advantage of providing medical care with a broader perspective than organ-based specialists and could, therefore, handle many different types of problems.

#### 3.1.3. Balance between Clinical Care, Education, and Research

The participants felt that general physicians, as opposed to organ-based specialists, saw providing student education as one of their jobs and were responsible and enthusiastic about it. Additionally, while organ-based specialists often conduct basic research, general physicians perform research that meets clinical and regional needs. The participants thought that this difference may be related to the general physician’s clinical practice, which considers the entire community and disease treatment.

One of the participants stated: 


*“I have the impression that general physicians provide student education fairly well. Other departments do not enthusiastically provide it; thus, I think it is a department that offers a balance between the three pillars of clinical care, education, and research”*
(Student 5).

The participants were attracted by the thought of becoming general physicians, as they felt that general physicians –unlike organ-based specialists—worked to maintain a balance between clinical care, education, and research.

#### 3.1.4. Meeting the Needs of Society

The participants stated that they often encountered the phrase “general medicine” and realized that the field had become more popular as the Japanese population had aged. They also believed that the demand for general physicians would increase to meet the needs of the times as the super-aging society would progress in the future.

One of the participants stated: 


*“I think that general physicians are popular because they appear on TV and in magazines. Maybe it is because the demand for healthcare has recently increased; I often hear that term [“general physician”] in society”*
(Student 8).

Another participant stated: 


*“We must reduce the number of specialists as much as possible, increase the number of hospitalists, and further increase the number of family physicians. Prevention and follow-up will be important in the future”*
(Student 1).

The participants stated that there is an increasing social demand for general medicine. Moreover, they stated that as the population ages, there should be an increase in the number of general physicians in charge of community medicine; this indicated that they considered the important role of general medicine in the society and that they looked forward to its further development in the future.

#### 3.1.5. Diversity and Development

The participants described that focusing on the novelty and diverse aspects of general medicine could lead to significant developments and ultimately improve the expertise of general physicians. The participants were also attracted by the prospect that general physicians could change their working style based on their workplace (e.g., hospitals or clinics).

One of the participants stated: 


*“It is better to mix it up and have more variety. Then, people who are truly interested in general medicine will participate in the field. I think that the definition is still ambiguous; there may be something we can do about this. From my perspective, it represents a stage where many people are doing various things because it is new”*
(Student 8).

The participants thought that focusing on the advantage of diversity in general medicine could lead to an expansion of the general physicians’ working style, and thereby in an increased number of medical students who would aspire to become general physicians. Thus, they believed that general medicine has a potential for further development.

### 3.2. Gaps between the Ideal and Reality of General Medicine

#### 3.2.1. Restrictions of Working Style

The participants had doubts concerning the continuity of patient care in general medicine; they believed that while general physicians exhibit a strong ability to diagnose difficult cases, their only role is to send the patients to appropriate departments after making a diagnosis and to ensure that the patients would be treated by an organ-based specialist. They mentioned that this perception may be influenced by the media and medical student study groups.

One of the participants stated: 


*“I have the impression that general physicians only make diagnoses, eventually sending patients to organ-based specialists. It appears that they mainly see undiagnosed cases in hospitals”*
(Student 7).

The participants who were interested in general medicine mentioned that general medicine education at their university and in study groups increased their interest in medicine and improved their ability to diagnose various conditions. Conversely, they believed that there was ambiguity regarding the scope of treatment intervention, and limitations regarding the working style of general physicians.

#### 3.2.2. Difficulty in Acquiring Skills

The participants admired general physicians’ vast knowledge and ability to diagnose a wide range of diseases. Conversely, they felt that general medicine was difficult and that not everyone could pursue it as continuing education. Finally, they believed that extraordinary abilities were needed to acquire these skills.

One of the participants stated: 


*“I think that general physicians need to learn too many things. If I try to achieve their skills as an organ-based specialist, I will end up learning nothing. I admire them, but I wonder if I can do it”*
(Student 10).

The participants worried that if they had specialized in general medicine, they would not expertise in any field as a result of trying to gain a broader knowledge. In addition, the medical students felt that general physicians face difficulty in acquiring skills and may need to continue studying. They mentioned that these factors would reduce the likelihood that medical students feel compatible with general medicine. They indicated that these issues would limit the development of general medicine as a field and, thus, they should be addressed.

#### 3.2.3. Ambiguity Regarding the Future

The participants expected the diversity of general medicine to develop; however, there was a stereotype that organ-based specialists comprised the majority of physicians. They believed that the future of general physicians, including their necessity, academics, and career paths, is unclear; therefore, they had concerns concerning their ability to specialize in organ-based medicine as general physicians because of the uncertain future of becoming a general physician.

Some of the participants stated: 


*“It has not been long since the program was established, so I am a little worried that there will not be many senior general physicians by the time I major, and I cannot clearly see the future [of general medicine]. I wonder whether or not it will be stable in the future. I think aiming for general medicine is a gamble”*
(Student 2).


*“I feel like general medicine is some kind of religion, in which people naively think that a concept such as empathy or holistic medicine is a wonderful thing without considering it on their own, or they blindly believe in what famous family doctors teach us. This ends up making general medicine self-centered”*
(Student 8).


*“Upon comparing papers published by a neurologist to papers published by a general physician, I am a little worried about how much it will affect the reader. Each has a different perspective and each has a high level of expertise. However, it [perspective] is complicated by the deeply rooted beliefs in medical culture wherein organ-based specialists are the majority and general physicians are the newcomers”*
(Student 6).

The participants worried about the lack of general medicine role models. Therefore, there was a lack of understanding that general physicians are improving patient outcomes and that there is a need for them. Moreover, they also worried regarding their future development, including academic evolution; hence, they pointed out that the unestablished future of general medicine might prevent medical students from aspiring to become general physicians.

#### 3.2.4. Unclear Expertise

The participants perceived general medicine—in many ways—based on the influence of their differing educational backgrounds and they could not define it as a specialization. They had a strong impression that there was a discrepancy between family physicians, who manage the health of the local community, and hospitalists, who manage entire hospitals and work as diagnosticians. They expressed discomfort in defining both as “general physicians”, making it difficult for them to understand the field of general medicine. They were confused by the difference between general medicine and general internal medicine as well as by the presence of primary care physicians in their local community. Moreover, they could not clearly distinguish between emergency and general physicians.

One of the participants stated: 


*“General physicians lack [role] coherence, which prevents us from understanding their specialty. There are physicians named ‘hospitalists’ and ‘family physicians’; they provide their own medical care in their own workplaces. Physicians, especially those who have been engaged in general medicine for a long time, tend to have their own practical style of general medicine. Students are likely to believe that the performance of the first general physician they encounter is the performance of all general physicians, causing them confusion when they encounter a different type of general physician”*
(Student 11).

The lack of unity in general medicine hindered the students’ understanding of the specialty. They were particularly worried that students would be biased, depending on the type of general physicians they met.

### 3.3. Factors Affecting Students’ Motivation for Specialization in General Medicine

#### 3.3.1. Current Clinical Situation of General Medicine

The participants argued that the university curriculum lacked appropriate education of and exposure to general medicine, which may limit their understanding and interest concerning general medicine. They mentioned that general medicine education at the university is biased, it is not impressive (as they only learn in the early years), and that they are not familiar with its content (as there are few general physicians at the university). They were also often exposed to criticisms of general physicians by organ-based specialists and were concerned that this might reduce motivation to pursue general medicine.

Some of the participants stated: 


*“Currently, there are not many people who promote general medicine as a specialty; taking that into account, I do not think there is much information regarding general medicine. I think the university should create more opportunities for general physicians to speak, even on a voluntary basis”*
(Student 4).


*“We do not have the opportunity to visit general physicians in clinical practice, (i.e., in community hospitals). I think that we have to increase exposure; it is about medical education on a national level”*
(Student 11).


*“When I have had the opportunity to talk to specialists, I was often told, “It is better to have one specialty”, or “You can do general medicine later”. I think it would be nice for medical students to learn how general physicians cooperate with specialists. After all, I have the impression that general physicians are disliked. It would be nice to hear of an incident from a specialist, such as: “It is amazing that the general physician introduced me to such a case””*
(Student 3).

Students’ understanding and interest concerning general medicine may be limited by the limited extent to which general medicine is taught at universities, the few opportunities to engage with the field, and criticism from organ-based specialists. The participants hoped that the environment for studying general medicine at universities and community hospitals would be adjusted.

#### 3.3.2. Lack of General Medicine Education in Universities

The participants highlighted the need for general medicine educators. They mentioned that they would like to understand general medicine concepts; especially, they would like to know what it is like working as a general physician (from the beginning of a medical career and for a long period of time thereafter) within this new medical specialty in Japan. However, education regarding a career in general medicine is lacking in medical school, and there are few general medicine educators. General medicine educators can provide medical students with appropriate knowledge of the medical care performed by general physicians (i.e., evidence-based medicine). Through this education, medical students can gain understanding concerning the importance of general medicine and the careers of general physicians, while continually improving their own individual medical skills and mastering medical education as a subspecialty.

Some of the participants stated: 


*“I think it is amazing that general physicians can manage patients with a wide range of diseases, but there are many people who think that they lack the specialized knowledge and skills. I believe that it is necessary for general physicians to further improve specialized medical skills so that their skills are not regarded as insufficient”*
(Student 5).


*“In the first place, doctors in medical schools are not trained to be educators. I think that there are doctors who consider teaching medical students, but whether the education would be effective is another matter. I believe that doctors do not know how to teach. That is why I want general physicians at community hospitals to work harder on medical education and to provide appropriate knowledge and skills for general medicine”*
(Student 9).

The participants suggested that as the number of physicians who have properly acquired general medicine skills increases, the number of junior physicians would increase. They argued that general physicians should take the lead in medical education because not all physicians are trained in it.

#### 3.3.3. Immaturity of Healthcare Policy

The participants mentioned that it was natural for medical students to avoid jumping into a new field, such as general medicine, because it has not been recognized as a specialty for very long. They also thought that improving the underlying medical policy was essential for the development of general medicine.

One of the participants stated:


*“I think that time will solve the problem. There are more and more general physicians now, aren’t there? I think that as the number of general physicians increases, more medical students will consider choosing general medicine as a career”*
(Student 2).

Another participant stated: 


*“I think it is possible to improve the career development of general physicians by adjusting medical policies. One idea is to limit the number of doctors in each region or specialty”*
(Student 1).

The participants were concerned that the novelty of general medicine and the immaturity of related healthcare policies may limit the increase in the number of general physicians. They thought that with time, general medicine may become more recognized as a career path. In addition, they thought that it would be necessary to revise healthcare policies to secure the increase in the number of general physicians.

#### 3.3.4. Large Regional Disparity

The participants stated that there may be regional differences between urban and rural areas relating to general medicine education, as some universities have general medicine departments, while others do not. They felt that the degree to which general medicine is promoted and the percentage of students interested in might differ from university to university.

One of the participants stated: 


*“I am wondering what the ratio is at other universities. I have the impression that Fukushima and Shiga are doing well, but in my hometown, Kumamoto, I hardly hear about general medicine. I wonder if it is different for universities in the city. One of my friends who wants to become a specialist said, “You are promoting general medicine because you are in Shimane, right?”*
(Student 3).

The participants were confused regarding learning and pursuing general medicine because of the different motivations that universities had relating to it.

#### 3.3.5. Ambiguity of the Chiiki-Waku System

The participants reported that there were discrepancies between government documents and the university’s Chiiki-Waku system policies regarding the defining roles of general physicians; thus, they were confused as to which one described the true role of a general physician. They felt that this made it difficult for them to understand general medicine. Students who are admitted via the Chiiki-Waku system are obligated to work in the prefecture after graduation. The participants had doubts regarding the way the g. Government and universities were trying to force students in the Chiiki-Waku system to become general physicians. They were concerned that, if the situation continues, the number of general physicians may increase, but the quality of their abilities would decline.

One of the participants stated: 


*“The general physicians provided by the Japanese Board of Specialists are quite like Doctor G [a diagnostician], but the Chiiki-Waku system is looking more for a family physician-type, which I have found to be confusing from the beginning”*
(Student 8).


*“First of all, I think that it would be better for the Government to eliminate the fixed model of general physicians, as Shimane will improve if all the students who are admitted via the Chiiki-Waku system become general physicians. I do not think it makes sense to just focus on the number of doctors. I am afraid of what is going to happen when this excitement about general medicine is over”*
(Student 8).

Varying information was provided regarding general medicine and there was confusion regarding the policies that were developed to secure the number of doctors in rural areas, such as the policies of the Chiiki-Waku system and the general medicine program to increase the number of general physicians. Therefore, the system added to the students’ confusion over general medicine and reduced their motivation for specializing in it.

## 4. Discussion

The research question of our study was: What prevents medical students from becoming general physicians? The results revealed three overarching themes in perceptions of the participating rural Japanese medical students regarding the future of general medicine: (1) hopes for the field of general medicine, (2) gaps between the ideals and reality of general medicine, and (3) factors affecting students’ motivation for specialization in general medicine. The medical students in this study expressed great hope for the future of general medicine and were particularly interested in the field. They had favorable impressions of it as community-based with a broad scope of practice and the potential for development. The gaps between these ideals and the reality of general medicine were identified. The immaturity of general medicine education at universities as well as that of relevant healthcare policies were revealed to be the causes of these gaps. The participants expressed great hope for the future of general medicine. It is necessary to construct an educational system based on their expectations while also considering the factors affecting their motivation for specialization in general medicine.

In a previous study, medical students from other countries had similar opinions regarding primary care and they improved their motivation to become primary care physicians through interactions with citizens in communities [22,23]; however, they also expressed negative opinions of it as being intellectually less challenging, boring, and of low prestige [22]. In contrast, the Japanese medical students in this study had positive impressions regarding general medicine. Interestingly, they expressed the following: (1) it provides a balance between clinical care, education, and research; (2) meets the needs of society; and (3) shows potential for diversity and development. These results may imply possible outcomes, such as the increase in the number of general physicians in countries with aging populations (i.e., Japan). Furthermore, primary care physicians have existed in Japan before training programs for general medicine were introduced in 2018. In studies conducted prior to 2018, primary care physicians were recognized as having broad knowledge and providing care to the local community, similar to how general physicians are perceived in the current study [22]. However, medical students believed that academics and research were more associated with organ-based medicine than with general medicine, and there was no mention of development [10,24]. Our results suggested that the establishment of a new department of general medicine may lead to higher expectations, allowing medical students to perceive it as a prestigious department with potential.

These results clarified the gap between the ideals and reality of general medicine as perceived by medical students who are engaged in the educational process; it highlighted issues for improvement and further possibilities for general medicine education in Japan and other countries. It was shown that while the working styles of general physicians are diversifying, general medicine education is not yet sufficiently developed; medical students were exposed to biased models of general physicians during the educational process. Similar to other countries, because of this bias, some students are under the impression that general physicians only utilize broad and superficial knowledge and lack specialized knowledge; they perceived that treatment is ultimately provided by organ-based specialists and not by general physicians [22,25]. Furthermore, the perception mentioned in this study that general medicine is difficult to master and not familiar to students, may reflect problems, such as academic ambiguity and the lack of role models. This may be a factor that prevents the promotion of general medicine as a specialization. Conversely, the negative opinions of medical students from other countries are opposed by the results of this study [21]. Japanese students recognize general medicine as an academically challenging field that stimulates the intelligence of medical students and residents. These issues are very important to consider to ensure that general physicians develop to the same degree as organ-based specialists. In the future, the value of general medicine should be demonstrated based on the available research findings, and not just based on its potential. Furthermore, to bridge these gaps we should clarify the actual learning process, to which Japanese general physicians are subjected; this clarification would help medical students gain a better understanding of general medicine. If general medicine education is improved in Japan as described above, it may provide a model for other countries that face a shortage of primary care physicians.

This study revealed the factors that create gaps between medical students’ ideals and the reality of general medicine. The participants strongly emphasized that current general medicine education at universities is poor and insufficient. Japan and other countries have reported that students can learn more regarding primary care and multi-professional cooperation in community hospital training; furthermore, they primarily learn concerning rare diseases (by organ) during university training [26,27]. The effectiveness of community-based general medicine education has been demonstrated [28,29,30]. In Japan, community-based medical practice became mandatory in the model core curriculum since 2016 [6]; however, while most universities have introduced community-based medical education [31], its quality has not been optimized [32]. The participants in this study felt that there were few opportunities to learn in community hospitals; therefore, to increase the number of general physicians, universities and community hospitals need to cooperate with one another to establish a curriculum, ensuring that students are fully exposed to and correctly understand general medicine at an early stage of their education.

We believe that it is particularly important to raise role models by appropriate training in general medicine, as the participants also mentioned that they could not learn from qualified general physicians. Moreover, the fact that they do not have role models in the field made it difficult for them to have a good understanding of general medicine. Furthermore, similar to medical students in other countries [8,12], we found that criticism toward general physicians from organ-based specialists had a negative impact on students’ perspectives; therefore, to enhance communication between organ-based specialists and general physicians, general physicians should strive to prove their ability to improve patient outcomes.

The results of this study highlighted the problems of the Chiiki-Waku system in Japan. The Chiiki-Waku system is a selective system of universities that offers scholarships to students, who are then obligated to work in the prefecture for a certain period of time after graduation [32]. Of the 9330 medical students enrolled in Japanese medical schools in 2020, 1679 (18%) were admitted to the Chiiki-Waku system [33]; this has significantly increased the number of physicians in areas experiencing shortages, thereby contributing to the improvement of physician maldistribution [31]. In our study, some students gained an interest in general medicine as a result of this system and felt a duty to pursue community medicine. Conversely, other students were concerned that the Government and the university would force them to be general physicians through this system. Although they were originally interested in general medicine, their enthusiasm might have waned because of pressure from the Government and the university. It is important to overcome this issue to increase the number of general physicians; thus, to maintain the development of general medicine, we should establish an autonomy-based educational system that will attract rather than compel students to specialize in general medicine.

Furthermore, medical students were confused regarding the differences between general and primary care physicians; in Japan, general, primary care, and family physicians have been providing primary care to local communities since before “general medicine” was recognized as a specialty. Additionally, patients in Japan can freely choose and visit any hospitals they wish, while physicians in any department can deliver primary care as primary care physicians [34]. This confuses medical students and results in misunderstandings regarding general medicine; therefore, clarifying the role of a general physician with pertinent information could eliminate this confusion, as well as increase students’ motivation to specialize in general medicine. Based on our findings, the relevant clinical experience can be vital for effective general medicine education. This education when delivered across community hospitals can motivate medical students and residents to become general physicians and increase their scope of practice [35,36]. Future studies should investigate the effect of community hospital-based clinical experience in general medicine on the motivation and learning of medical trainees in various contexts.

This study had certain limitations. First, the participants were limited to medical students at Shimane University; however, as aforementioned, the number of general physicians in Japan is small. In addition, standardized education is provided according to the core model curriculum established by the Ministry of Health, Labor, and Welfare, as well as global standards for medical education. Therefore, the transferability of this study was maintained. In contrast, our results suggested that there are regional differences in general medicine education; hence, similar studies in other regions may reveal differing issues. Second, our findings were based on a single university in Japan. However, as previously stated, Japan has the most aging population worldwide. Similarly, the demand for primary care is expected to increase in many other countries with the aging society. Therefore, the trajectory of the establishment of general medicine as a specialty in Japan is valuable and may be transferred to other countries. Finally, the interviewers and participants were medical students. However, we succeeded in grasping the essence of the students’ views, as the interviewers and participants had interacted before the study, allowing the participants to express their views freely and easily. In contrast, the interviewer was a medical student who had not practiced general medicine; therefore, some of the questions posed might not have sufficiently considered the actual clinical setting.

## 5. Conclusions

Our study revealed that medical students have high expectations toward general medicine and its potential for future development. In contrast, they acknowledge that there is a gap between the ideals and reality of general medicine (such as the fact that the specialty of general medicine is ambiguous). In addition, the factors that influence the medical students’ perceptions of general medicine were clarified, including the lack of exposure to general medicine and immaturity of general medicine education and relevant policies. For the further development of general medicine education, it is necessary to revise the curriculum of general medicine and medical policies based on these perceptions.

## Figures and Tables

**Table 1 healthcare-09-01256-t001:** The six-year general medicine curriculum of Shimane University Faculty of Medicine as of 2019.

Year	1st	2nd	3rd	4th	5th	6th
Lecture	Introduction to health science		Environmental health and preventive medicine			
	Community medicine					
	Seminars of community medicine (optional)					
PBL			General medicine (including classroom lectures)			
Clinical practice	Community-based practice (optional)				General medicine (at university hospital)	
					Community-based practice (at community hospital)	
	Optional practice					

**Table 2 healthcare-09-01256-t002:** Three themes and their concepts using thematic analysis.

Theme	Concept
Hopes for the field of general medicine	Community-based medicine
	Broad scope of practice
	Balance between clinical care, education and research
	Meeting the needs of society
	Diversity and development
Gaps between the ideal and reality of general medicine	Restrictions of working style
	Difficulty in acquiring skills
	Ambiguity regarding the future
	Unclear expertise
Factors affecting students’ motivation for specialization in general medicine	Current clinical situation of general medicine
	Lack of general medicine education in universities
	Immaturity of healthcare policy
	Large regional disparity
	Ambiguity of the Chiiki-Waku system

## Data Availability

The datasets used and/or analyzed during the current study available from the corresponding author on reasonable request.

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
