# Peer review of "The Perception of Rural Medical Students Regarding the Future of General Medicine: A Thematic Analysis"

_healthcare, 2021, doi:10.3390/healthcare9101256_

Round 1

Reviewer 1 Report

We are grateful for the effort and commitment you are showing with the article. The improvements that were pointed out to you in previous reviews are noticeable.

However, despite all the changes made, the quality of the article is not sufficient to be published in this journal. 

Even so, I encourage you not to leave it aside, and to submit it to another journal where it may fit in MDPI editorial.

Author Response

Reviewer 1

​​We are grateful for the effort and commitment you are showing with the article. The improvements that were pointed out to you in previous reviews are noticeable.

However, despite all the changes made, the quality of the article is not sufficient to be published in this journal.

Even so, I encourage you not to leave it aside, and to submit it to another journal where it may fit in MDPI editorial.

Response:         

The authors would like to thank the reviewer for his/her constructive critique to improve the manuscript. We have made every effort to address the issues raised and to respond to all comments.

We strongly believe that our article could be published in Healthcare. Please note that we have submitted it for the special issue "Innovations in Ecological Public Health and Health Education." According to the information found on the journal’s website regarding its scientific interests, innovative researches concerning medical education and curriculum are considered for publication [1]. Therefore, we believe that our manuscript is appropriate for this journal, as we have revealed the perception of medical students regarding the future of general medicine. In addition, we have highlighted several issues that need to be improved in medical education and curriculum.

We would like to ask the reviewer to re-consider our manuscript for publication in Healthcare.

Reference

[1] Healthcare (n.d.). Retrieved September 11, 2021 from https://www.mdpi.com/journal/healthcare/special_issues/Ecological_Health_Education#published

Reviewer 2 Report

The article is improved in content and aspect. I still consider the introduction too long but if the authors considered it necessary could be accepted.

The scientific style used, with many quotes, is unusual to my scientific editing experience. But again, if the authors keep the very essential of the information, the readability could be better.

In conclusion, I propose minor revision for the style, keeping the moon scientific data at the minimum. 

Author Response

Reviewer 2

The article is improved in content and aspect. I still consider the introduction too long but if the authors considered it necessary could be accepted.

The scientific style used, with many quotes, is unusual to my scientific editing experience. But again, if the authors keep the very essential of the information, the readability could be better.

In conclusion, I propose minor revision for the style, keeping the moon scientific data at the minimum.

Response:         

We would like to thank the reviewer for evaluating our manuscript and for his/her comments. Please note that we have reduced the number of words in the revised manuscript. Especially, we have improved the expression and removed the unnecessary parts. Additionally, we have removed some quotes, as per the reviewer’s suggestion.

Reviewer 3 Report

I would like to thank the authors of the publication and the study for presenting a very interesting paper. The value of the publication is both: the topic related to the undervalued general medicine by medical students worldwide and the qualitative analysis conducted, thanks to which it is possible to learn not only about attitudes but also to penetrate into the nature of motivation of social players, in this case medical students.

In my opinion, the paper would have gained more value after clarifying several methodological elements related to the implementation of the study.

111 It is not entirely clear to me how many students are studying at the Faculty of Medicine and to what number invitations to the study were sent.

124 How exactly were the students selected for the study, was it the top 12 or were they selected according to other categories?

128 In an instrument that is semi-structured in qualitative research, researchers have a set of questions that they must ask each respondent. Was this the case in this study? How many of these questions were there? It may be worth clarifying what the nature of a semi-structured interview is.

129-131 Were these the only questions that were asked of all respondents? Were spontaneous responses waited for? Were follow-up questions asked of the respondents?

132 What does it mean that the questions were updated? Did they change with each respondent?

Given the range of doubts and ambiguities above, it may be worthwhile in the Material and Methods chapter to highlight the “Interview tool” or “Interview scenario” subsection and describe the tool that was used to conduct the study?

544 "We asked the question:...." I understand that was a general research question, not a question that was asked of each student in the questionnaire?

Author Response

Reviewer 3

​​I would like to thank the authors of the publication and the study for presenting a very interesting paper. The value of the publication is both: the topic related to the undervalued general medicine by medical students worldwide and the qualitative analysis conducted, thanks to which it is possible to learn not only about attitudes but also to penetrate into the nature of motivation of social players, in this case medical students.

In my opinion, the paper would have gained more value after clarifying several methodological elements related to the implementation of the study.

Response:

The authors would like to thank the reviewer for his/her constructive critique to improve the manuscript. We have made every effort to address the issues raised and to respond to all comments. Please, find next a detailed, point-by-point response to the reviewer's comments.

111 It is not entirely clear to me how many students are studying at the Faculty of Medicine and to what number invitations to the study were sent.

Response:

In accordance with the reviewer's comment, we have revised this part as follows:

“There were 696 medical students at Shimane University at the time of study [18]. We posted the information on social networking services which almost all of them used and recruited those who were interested in general medicine. As of 2019 (before the COVID-19 pandemic), the participants had learned concerning general medicine based on the Shimane University Faculty of Medicine curriculum.” (Lines 97–101)

124 How exactly were the students selected for the study, was it the top 12 or were they selected according to other categories?

Response:

We would like to thank the reviewer for the question. When performing qualitative researches, it is possible to conduct a purposeful sampling strategy according to the research questions [1]. In this study, the first author unilaterally selected the participants who were expected to present a variety of opinions regarding general medicine in order to obtain as much opinions as possible. Then, we tried to generalize the results by describing the participants in detail to increase the validity of the results. We did not select the top 12 students.

128 In an instrument that is semi-structured in qualitative research, researchers have a set of questions that they must ask each respondent. Was this the case in this study? How many of these questions were there? It may be worth clarifying what the nature of a semi-structured interview is.

129-131 Were these the only questions that were asked of all respondents? Were spontaneous responses waited for? Were follow-up questions asked of the respondents?

Response:

We agree that this point requires clarification. Therefore, we have revised the corresponding part in the text as follows:

“As a guided interview for the semi-structured interviews, we first asked all participants three questions: "What do you think of general medicine?", "What prevents you from specializing in general medicine?", and "What should be improved regarding general medicine?" Following these questions, the interviewer added questions and expanded the topics according to the answers to understand the participants' real perceptions. The interviewer and the coauthor who is a general physician analyzed and discussed the data after each interview. Then, we added any unclear and newly emerging questions in addition to the three questions. We continued to collect and analyze the data until the data were saturated. We performed 12 interviews in total.”  (Lines 114–122)

132 What does it mean that the questions were updated? Did they change with each respondent?

Given the range of doubts and ambiguities above, it may be worthwhile in the Material and Methods chapter to highlight the “Interview tool” or “Interview scenario” subsection and describe the tool that was used to conduct the study?

Response:

We agree that this point requires further clarification. Please note that we have analyzed and discussed the data after each interview, and added any unclear and newly emerging questions in addition to the three questions in the set. We have revised the corresponding part in the manuscript as follows:

“As a guided interview for the semi-structured interviews, we first asked all participants three questions: "What do you think of general medicine?", "What prevents you from specializing in general medicine?", and "What should be improved regarding general medicine?" Following these questions, the interviewer added questions and expanded the topics according to the answers to understand the participants' real perceptions. The interviewer and the coauthor who is a general physician analyzed and discussed the data after each interview. Then, we added any unclear and newly emerging questions in addition to the three questions. We continued to collect and analyze the data until the data were saturated. We performed 12 interviews in total.“ (Lines 114–122)

544 "We asked the question:...." I understand that was a general research question, not a question that was asked of each student in the questionnaire?

Response:

We would like to thank the reviewer for the question. Please note that this was a research question for the study itself. We have revised the corresponding part in the manuscript as follows: “The research question of our study was: What prevents medical students from becoming general physicians?” (Lines 485-486)

[1]Patton, Michael Quinn. Qualitative research & evaluation methods: Integrating theory and practice. Sage publications, 2014.